

# GLOFRIM v1.0 – A globally applicable computational framework for integrated hydrological-hydrodynamic modelling

Jannis M. Hoch[1,2], Jeffrey C. Neal[3], Fedor Baart[2], Rens van Beek[1], Hessel C. Winsemius[2,4], Paul D. Bates[3], Marc F.P. Bierkens[1,2]

[1] Department of Physical Geography, Utrecht University, P.O. Box 80115, 3508 TC Utrecht, the Netherlands
[2] Deltares, P.O. Box 177, 2600 MH Delft, the Netherlands
[3] School of Geographical Sciences, University of Bristol, University Road, Bristol, BS8 1SS, UK
[4] Institute for Environmental Studies, VU University, De Boelelaan 1087, 1081 HV, Amsterdam, the Netherlands

*Correspondence to*: Jannis M. Hoch (j.m.hoch@uu.nl)

**Abstract.** To increase the representation of physical processes in inundation modelling, current research approaches aim to integrate both hydrological and hydrodynamic models. A previous study by Hoch et al. (2017) showed that spatially explicit coupling approaches can outperform stand-alone runs by single-purpose models as they combine spatially distributed model forcing by hydrological models with more sophisticated routing schemes in hydrodynamic models. We here present
GLOFRIM, a globally applicable computational framework for integrated hydrological-hydrodynamic modelling, to facilitate such coupling approaches and to cater for an ensemble of models to be coupled. It currently allows for coupling the global hydrological model PCR-GLOBWB with either Delft3D Flexible Mesh (DFM), solving the full shallow-water equations and allowing for spatially flexible meshing, or LISFLOOD-FP (LFP), solving the local inertia equations and running on regular grids. The main advantages of the framework are its open and free access, its global applicability, its
versatility, and its extensibility with other hydrological or hydrodynamic models. Before applying GLOFRIM to an actual test case, we benchmarked both DFM and LFP for a synthetic test case. Results show that for sub-critical flow conditions, discharge response to the same input signal is near identical for both models, which agrees with previous studies. We subsequently applied the framework to the Amazon River basin to test the framework thoroughly and, in addition, to perform a first-ever benchmark of flexible and regular grids at the large-scale. Both DFM and LFP produce comparable results in
terms of simulated discharge with LFP exhibiting slightly higher accuracy as expressed by a Kling-Gupta-Efficiency of 0.82 compared to 0.76 for DFM. However, benchmarking inundation extent between DFM and LFP over the entire study area, a critical success index of 0.46 was obtained, indicating that the models disagree as often as they agree. Differences between models in both simulated discharge and inundation extent is to a large extent attributable to the gridding techniques employed. In fact, the result show that the numerical scheme of the inundation model and the gridding technique can
contribute as strongly to deviations in simulated inundation extent as, unlike the global flood model inter-comparison by Trigg et al. (2016), we control for model forcing and boundary conditions. This study shows that the presented computational framework is robust and widely applicable. GLOFRIM is designed as open access and to be easily extendable, and thus we hope that other large-scale hydrological and hydrodynamic models will be added, eventually capturing more locally relevant processes as well as allowing for more robust model inter-comparison, benchmarking, and
ensemble simulations of flood hazard at the large scale.





## 1 Introduction

In the latter half of the last century, losses due to riverine floods increased greatly, leading to economic losses of more than $1 billion and 220,000 casualties since 1980 (Munich Re, 2013; Visser et al., 2012). Much of this increase is thought to be due to continued settlement along rivers and shifts in climate patterns, meaning that this tendency will most likely be exacerbated in the future (Ceola et al., 2014; Hirabayashi et al., 2013; Winsemius et al., 2016). Sound inundation estimates are therefore paramount to enhance our process understanding and to provide better flood hazard estimates for risk models. Since recent research showed that flood inundation can easily affect large areas, in particular neighbouring river basins (Jongman et al., 2014), it is vital that flood hazard models can simulate the relevant processes over large domains. Applying such large-scale models has the additional advantage of facilitating the identification of risk hotspots and providing critical insight into data-scarce areas (Ward et al., 2015). In fact, there are already a number of global-scale inundation models available (Dottori et al., 2016; Pappenberger et al., 2012; Sampson et al., 2015; Winsemius et al., 2013; Yamazaki et al., 2011), differing in their process descriptions and computational engine. While some approaches derive flood hazard from a coarse-scale hydrological model and subsequent downscaling, others force fine-scale hydrodynamic models with globally regionalized discharge data. A first inter-comparison of global flood hazard models by Trigg et al. (2016) for the African continent, however, revealed that they agree for only 30%-40% of aggregated flood extent, thus indicating that the representativeness of local flood risk estimates may depend strongly on the computational engine opted for as well as on the model forcing applied. Identifying the exact reasons for model disagreement was impossible due to the diversity of methods and lack of a systematic approach to the inter-comparison where individual aspects of the modelling frameworks could be isolated.

Employing a global hydrological model (GHM) such as PCR-GLOBWB (van Beek et al., 2011; van Beek and Bierkens, 2008), WaterGAP (Alcamo et al., 1997; Döll et al., 2003) or VIC (Liang et al., 1994; Wood et al., 1992) has the benefit of providing spatially distributed surface runoff and routed discharge simulations, thereby facilitating direct forcing for spatially distributed inundation models. In addition, these models are usually forced by global meteorological data, hence diminishing the dependency on observed data as well as allowing for easier implementation of future climate scenarios. However, the routing schemes currently implemented in large-scale hydrological models can generally be described as simplistic as they are based on gridded drainage networks at coarse spatial resolution, with the currently finest spatial resolution of GHMs being 5 arcmin or around 10 km x 10 km at the Equator (Bierkens, 2015). Furthermore, discharge accuracy may be reduced in low-gradient catchments since topography at this scale is generally parameterized in distribution functions and river routing is often represented by a simple scheme, such as the kinematic wave approximation.

Hydrodynamic models, on the other hand, can be built in numerous ways for inundation modelling, typically in 1-D, 2-D or combined 1-D/2-D, and are mostly forced with gauged discharge data or synthesized flood waves. While such approaches do not require rainfall-runoff conversion, they are problematic for studies concerning large-scale climate change impacts or the seamless simulation of flood events and their spatial correlation (Jongman et al., 2014). Some models like CaMa-Flood (Yamazaki et al., 2011) route a priori computed hydrology-based surface runoff with 1-D hydrodynamics and parameterized 2-D floodplain storage. Applying such a 1-D/2-D approach, however, does not allow for explicit modelling of floodplain flow pathways as well as channel-floodplain interactions. Explicitly representing these processes would be beneficial as they are known to greatly influence inundation dynamics and patterns (Trigg et al., 2009). Compared to hydrological models, hydrodynamic models solving the full SWE or at least a more advanced approximation such as the local inertia equations (LIE) have the advantage of providing a better representation of backwater effects, which are important flood-triggering processes (Meade et al., 1991; Moussa and Bocquillon, 1996; Paiva et al., 2013). Another difference to GHMs is that current





applications of hydrodynamic models at the large to global scale can run at spatial resolutions of up to 1 km (Sampson et al., 2015), greatly facilitating the representation of both relevant channel-floodplain interactions (Rudorff et al., 2014a, 2014b) and flow pathways on floodplains (Rudorff et al., 2014a; Tayefi et al., 2007) as well as enhancing the usability for decision-making processes (Beven et al., 2015; Trigg et al., 2016). Notwithstanding these advantages, hydrodynamic models lack an

advanced implementation of hydrological processes and thus may overpredict both inundation extent and depth as, for instance, groundwater infiltration and evaporation from inundated floodplains are currently not fully accounted for.

Large scale flood hazard estimates may thus benefit from increased integration of hydrology and hydrodynamics in inundation models to allow for physically more integrated assessments and to compensate for their respective shortcomings. In fact, hydrological-hydrodynamic coupling was already applied in a number of studies (Biancamaria et al., 2009; Lian et

al., 2007; Schumann et al., 2013), but none of these studies coupled hydrology and hydrodynamics in a spatially explicit manner, that is on a grid-by-grid basis. Instead, they employed output from hydrological or land-surface models as input to the 1-D/2-D hydrodynamic model LISFLOOD-FP at a number of locations (Bates et al., 2010; Bates and de Roo, 2000). While such approaches reduce the dependency on gauged data or synthesized flood waves, they cannot fully account for important and spatially distributed hydrological flood-triggering processes within the model domain. This would, however,

be advantageous to support the assessment of spatial correlations of flood waves in adjacent river basins, which are shown to increase trans-national flood risk (Jongman et al., 2014). A further valuable contribution for promoting the coupling of models from different disciplines was realized by the Community Surface Dynamics Modelling Systems group (CSDMS) with their development of the Web Modelling Tool (WMT; CSDMS (2017)). This tool enables the user to create a coupled model from a list of readily available models and run it on a server of CSDMS. Whilst this is an important step towards

integrated modelling between disciplines, applicability is hampered by the fact that model code is not openly accessible and that the number of available models is limited and predefined.

Recently, Hoch et al. (2017) coupled PCR-GLOBWB (hereafter PCR) with the hydrodynamic model Delft3D Flexible Mesh (hereafter DFM; Kernkamp et al. (2011)) for the Amazon River basin to integrate the hydrological and hydrodynamic processes occurring over the entire study area. Results indicate that spatially explicit coupling of hydrological and

hydrodynamic models can improve the representation of inundation for all river reaches, not only those that are connected to upstream boundary conditions. Findings also corroborate that spatially distributed forcing retrieved from a hydrological model in combination with a sophisticated river routing scheme outperforms results obtained with both models run in stand-alone mode.

Even though these results are promising, it has to be acknowledged that the accuracy of a hydrological and hydrodynamic

model can vary strongly, depending on the chosen study area, model parameterization, model structure, numerical scheme or the use of different input data (Li et al., 2015; Trigg et al., 2016). It would hence be advantageous to base the choice of the coupled models on their local performance, potentially outperforming predefined set-ups, or simply on the model schematization at hand.

To facilitate such model selection and to further promote the coupling of large-scale hydrological and hydrodynamic models,

we developed GLOFRIM, a GLObally applicable computational FRamework for Integrated hydrological-hydrodynamic Modelling. In addition to the work of Hoch et al. (2017), it includes the widely used hydrodynamic model LISFLOOD-FP (hereafter LFP; Bates and de Roo (2000)) and an improved as well as extended coupling algorithm, thus catering a wider range of model schematizations and applications. As we believe that by combining the locally best-performing hydrological and hydrodynamic models can better capture all relevant processes, GLOFRIM is designed in an expandable way to



eventually incorporate more models. Furthermore, the framework is openly available under GNU 3.0 license[1] to stimulate collaboration and idea exchange within the scientific community. Key assets of the framework are its free and open accessibility, its global applicability, its versatility, and its potential to be further developed to a full two-dimensional coupling scheme between hydrology and hydrodynamics, which would play a particularly crucial role in basins in semi-arid

climates as for instance the Niger (Dadson et al., 2010; Mahe et al., 2009).

In the remainder of the paper, we first describe the model components of the framework and thereafter the framework and its functionalities in detail. Subsequently, we compare the two hydrodynamic models in a simple synthetic test case to obtain a first understanding of possible differences, in particular in terms of their numerical schemes. As means for benchmarking, we assess simulated discharge along the flow paths as well as run times for a 1-D and 2-D set-up individually. We then apply

GLOFRIM to one-directionally couple PCR with both DFM and LFP and benchmark the set-ups for an actual test case in the Amazon River basin, hence also constituting a first comparison of flexible and regular grids for large-scale applications. For model benchmarking, we assess simulated discharge, water levels, run times, and inundation extent. Pearson's correlation r, the root mean square error RMSE, and the Kling-Gupta-Efficiency KGE (Gupta et al., 2009) are determined by comparison to observed discharge data from the Global Runoff Data Centre (GRDC) at Óbidos. We opt for GRDC data as the presented

approach is merely based on input data sets with global coverage. Simulated water levels are compared at an upstream, midstream, and downstream station to assess a) whether water level dynamics are correctly represented and b) to what extent DFM and LFP differ or agree in their water level computations. Computational efficiency is assessed by comparing the run times of the coupled set-ups. To benchmark inundation extent from DFM with LFP, we determine the hit rate H, false alarm ratio F, and the critical success index C based on inundation maps of both models at the end of the simulation. No validation

of simulated inundation extent was performed as Hoch et al. (2017) already showed good agreement of results obtained with DFM for the same study domain.

This openly available computational framework makes a valuable contribution to current inundation modelling at the large scale by enhancing the integration of hydrological and hydrodynamic model processes, which eventually may lead to improved decision making as well as planning of adaption and mitigation measures.

## 2 Models

Currently, GLOFRIM includes the hydrological model PCR-GLOBWB as well as the hydrodynamic models Delft3D Flexible Mesh and LISFLOOD-FP. Hereafter, an overview of the main features of the models is provided. For further details regarding model development and model set-up, we refer to the specific manuals or websites.

### 2.1 PCR-GLOBWB

To generate hydrological input, the global hydrological model PCR-GLOBWB (PCR) is currently incorporated in the framework. It can be applied at 30 arcmin resolution (approximately 55 km x 55 km at the Equator) as well as at 5 arcmin resolution (approximately 10 km x 10 km at the Equator), which may increase accuracy but also runtime. PCR is entirely coded in PCRaster Python (Karssenberg et al., 2010) and distinguishes between two vertically stacked soil layers, an underlying groundwater layer, and a surface canopy layer. Water can be exchanged vertically, and excess surface water can

be routed horizontally along a local drainage direction (LDD) network employing the kinematic wave approximation. The model is forced with Climate Research Unit (CRU) precipitation and temperature data (Harris et al., 2014), and evaporation

---

[1]  The code and user manual of GLOFRIM is downloadable at doi.org/10.5281/zenodo.597107





is computed using the Penman-Monteith equation. Data sets are downscaled to daily fields for the period from 1957 to 2010 using ERA40/ERAI (Kållberg et al., 2005; Uppala et al., 2005). Besides, PCR is able to account for domestic and industrial water consumption by accounting for water demand data (FAO, 2017). For more information on PCR, we refer to van Beek and Bierkens (2008) and van Beek et al. (2011). PCR was already applied for a wide range of studies such as flood and

drought forecasting (Yossef et al., 2012), human impact on droughts (Wanders and Wada, 2015), global water stress (van Beek et al., 2011), and global groundwater simulations (de Graaf et al., 2015). More relevant to this study, PCR constitutes the computational backbone of the "GLObal Flood Risk with IMAGE Scenarios" framework (GLOFRIS; Winsemius et al., (2013)) which is also used as basis for the Aqueduct Global Flood Analyzer of the World Resources Institute (World Resources Institute, 2017).

**2.2 Delft3D Flexible Mesh**

Delft3D Flexible Mesh (DFM) allows the user to schematize the model domain with a flexible mesh in 1-D/2-D/3-D, and therefore supports the computationally efficient schematization of topographically challenging areas such as river bends or irregular slopes. The model solves the full Saint-Venant equations, or shallow-water equations (SWE). The main partial differential equations solved by DFM are

$$\frac{\partial h}{\partial t} + \nabla \cdot \left( h\mathbf{u} \right) = 0 \tag{1}$$

$$\frac{\partial \mathbf{u}}{\partial t} + \frac{1}{h}\left( \nabla \cdot (\mathrm{h}\,\mathbf{u}\mathbf{u}) - \mathbf{u}\nabla \cdot \left( h\mathbf{u} \right) \right) = -g\nabla \zeta + \frac{1}{h}\nabla \cdot \left( vh\left( \nabla \mathbf{u} + \nabla \mathbf{u}^{T} \right) \right) + \frac{1}{h}\frac{\tau}{\rho} \tag{2}$$

With

$$\nabla = \left( \frac{\partial}{\partial x}, \frac{\partial}{\partial y} \right)^{T} \tag{3}$$

$\zeta$ being the water level, $h$ the water depth, $\boldsymbol{u}$ is the velocity vector, $g$ the gravitational acceleration, $v$ the viscosity, $\rho$ the water mass density, and $\tau$ the bottom friction. For 1-D flow, the equations remain the same except that the viscosity $v$ does not contain horizontal eddy viscosity. For further technical details and derivation, we refer to the Technical Manual (Deltares, 2017a). DFM is an openly accessible model and can be obtained by contacting Deltares (https://www.deltares.nl/en/software/delft3d-flexible-mesh-suite/). Besides riverine flood hazard modelling, it also caters a

wider range of applications, for instance groundwater flow, sediment transport, and water quality simulations in 1-D, 2-D, and 3-D. For more information regarding the application of DFM, we refer to the User Manual (Deltares, 2017b). Due to its very recent publication, only a limited number of published studies using DFM are available. It was, for instance, applied in a global-scale reanalysis for extreme sea levels (Muis et al., 2016). In another study, Castro Gama et al. (2013) applied DFM to model flood hazard at the Yellow River, and concluded that applying a flexible mesh reduces computation time by a

factor 10 compared to square grids with equal quality of model output.

**2.3 LISFLOOD-FP**

LISFLOOD-FP (LFP) is a widely used, raster-based model to compute floodplain inundation. Since its first version (Bates and de Roo, 2000), it has regularly been adapted and improved (Bates et al., 2010), for instance by adding a sub-gridding scheme to account for channel flow within cells (Neal et al., 2012b).





It is possible to run LFP with different set-ups: a 2-D only, a 1-D, a 1-D/2-D or a sub-grid model, with the latter being the most accurate for large-scale inundation modelling approaches as it greatly increases floodplain connectivity (Neal et al., 2012b).

When using the sub-grid scheme, LFP solves the subsequent equation for channel flow that is based on a simplification of

the SWE ignoring advection (Bates et al., 2010; Neal et al., 2012b). Here $q$ denotes the flow per unit width, $g$ the gravitational acceleration, $\zeta$ the water level, $R$ the hydraulic radius, $n$ Manning's surface roughness, and $\nabla$ the gradients in x- and y-direction as described in Eq. 3:

$$\frac{\partial q}{\partial t} + \nabla g h \zeta + \frac{g n^2 q^2}{R^{4/3} h} = 0 \qquad (4)$$

Mass conservation is implemented as

$$\nabla(h + q) = 0 \qquad (5)$$

Whereby $\Delta t$ denotes the time step, $\Delta x$ the cell size and $i,j$ the cell indices. For further information about model development, derivation of numerical solutions, assumptions, and validations, we refer to the above-mentioned papers.

LFP is specifically developed to model floodplain inundation and has been used in a wide range of studies. Most notable in the context of large-scale flood hazard modelling is the work by Sampson et al. (2015) who applied LFP to compute global

estimates of flood hazard and risk as well as by Schumann et al. (2013) and Biancamaria et al. (2009) who used LFP to simulate inundation in the Zambezi River and Ob River, respectively, forced with lateral input from a land surface model.

The BMI adapter (see subsequent section) was implemented for LFP version 5.9 which provides all relevant features, in particular the sub-gridding scheme, to model large-scale inundation.

**2.4 Basic Model Interface**

Generally, the BMI has several functions that can be called from external applications like, as in this case, a Python script. To make these functions available for a model, a BMI adapter needs to be developed for each model with respect to the specific internal model structure and programming language. Whilst PCR is already written in Python and its BMI implementation is hence straightforward, DFM offers a native C-compliant BMI-implementation. For LFP, which is written in C++, the code and file structure had to be slightly adapted to agree with the requirements for the BMI. Once a BMI

adapter is developed, it is possible to execute a set of functions: first, the user can initialize the models by using the BMI adapter. Second, the BMI adapter allows for retrieving a number of variables from memory. This number exposed through the BMI adapter can be defined during the development of the BMI adapter and is thus not limited to a pre-set range. Third, the manipulated variables can be set back to the original model or can be used to overwrite variables in one or multiple other models, given that they agree to the internal data structure of those models. Fourth, models connected to a BMI adapter can

be updated at a user-specified time step, hence enabling online-coupling of models. In this way it is possible to get, change, and set variables during the execution of the models in use on a time step basis. Last, models can be finalized to end the computations. It is noteworthy that implementing the BMI functions does not alter any functionality or routines in the models. Both DFM and LFP, although not being coded in Python, can be called from within Python using the BMI-python package (see https://github.com/openearth/bmi-python). For further information regarding the BMI, we refer to Peckham et

al. (2013) and the related website (CSDMS, 2016).





## 3 The computational framework GLOFRIM

The computational framework presented here consists of two key elements, a) the actual code and b) a settings-file. Hereafter, a brief overview is given of their main properties. More detailed information and outline is provided in the files themselves.

The computational backbone of GLOFRIM is entirely written in Python 2.7 and was developed and tested on Ubuntu systems. By means of a python-file ("couplingFramework_v1.py" in the downloadable data), the steps for model coupling are executed (see Figure 1 for a flow chart). The models are first initialized, that is, the model configuration files of each model are read and the internal steps required to obtain an initial state of the models are prompted by the BMI adapter. Thereafter, the BMI adapter is used to retrieve all required model variables, especially geometry information. This

information is subsequently used to construct the grids of the models and to spatially couple them by overlay and grid-to-grid assignment. A many-to-one assignment based on raster indices is performed and the routing computations in PCR are turned off for all cells signalled as coupled. In case no 1-D or 2-D hydrodynamic cells are located within a PCR cell, this cell is therefore not considered to be coupled and the routing scheme as implemented in PCR prevails. Further information about the spatial coupling can be found in Hoch et al. (2017). Once the models are spatially coupled, the update loop commences.

During execution of this loop, PCR will be updated at each time step – typically one day –, and surface runoff and discharge output will be retrieved as well as externally adapted to agree with the data structure of the chosen hydrodynamic model. Subsequently, either the water depth or a flux variable in the hydrodynamic model will be overwritten, and finally the hydrodynamic model will be updated until it reaches the same simulation time as PCR. The loop is exited once a user-specified number of time steps is reached. It should be noted that in the current version of the framework, only one-

directional coupling from hydrology to hydrodynamics is supported, possibly leading to local overprediction of simulated discharge as there is, for instance, no re-infiltration of water going overbank. Future research will thus focus on extending this to a full two-directional coupling scheme with feedback loops from hydrodynamics to hydrology. Such two-way coupling would, for instance, contain explicit modelling of hydrological processes over inundated areas in the hydrodynamic model.

To specify all relevant information about the coupling run to be performed, a configuration file is needed ("default.ini" in the downloadable data). Besides all critical paths to model data, other model settings can be defined in the configuration file, for example the number of model time steps. In general, settings defined in the ini-file overrule those specified for the individual models. In the current version of GLOFRIM, three options need to be specified to realize model coupling: by activating the so-called "River-Floodplain-Scheme", by specifying the variables to be updated, and by choosing for hydrodynamic models

in either spherical or projected coordinate systems.

The so-called "River-Floodplain-Scheme" (RFS) defines where output from PCR is coupled to. If RFS is activated, water volume is directly coupled to the 1-D channels of the hydrodynamic model while, when RFS is inactive, water is distributed over all grid cells of the 2-D domain. Applying the RFS has two major advantages: first, it reduces run times as data exchange and computations need to be performed for a smaller number of cells; second, using RFS in large-scale

applications with sufficient channel information reduces the dependency on the accuracy of the 2-D elevation data which is known to contain strong vertical bias, in particular when derived from remotely sensed global data sets such as Shuttle Rader Topographic Mission (SRTM) data. In particular simulation of flow over vertically irregular terrain resulting in super-critical regimes is contra-indicated for LFP because of its use of the LIE. In case overland flow needs to be modelled by LFP, we advise to take measures accordingly, for instance by limiting flow velocities. For DFM we found that runs are more stable,

yet slower, when deactivating the RFS.





Second, it is possible to force the hydrodynamic models by updating the water depth variable in *m* or by updating fluxes, which are expressed as discharge in LFP in $m^3/s$ and as precipitation in *mm/d* in DFM. For DFM, added daily water depth is divided over a number of user-specified time steps, hence reducing the computational load, while fluxes are daily constants. We found that updating fluxes reduces run times compared to states, and hence advise opting for for this option. While it is

also possible to perform state-updating in LFP, we found that this option has to be used carefully as it easily increases run times. This is because it is currently not possible to update LFP at a user-specified time step due to the Courant-Friedrichs-Lewy condition. It may hence happen that gradients between added daily water depths are too steep, increasing the risk of model instability. We therefore recommend applying flux-updating in LFP instead.

Third, it is possible to use the hydrodynamic models with Cartesian coordinates, although PCR runs in spherical coordinates.

By providing the projected coordinate system the model is based on, the computational framework can translate the grid into spherical coordinates and perform the grid overlay and cell assignment, thus guaranteeing the applicability of all already existing hydrodynamic schematizations. All other computations remain unaffected by the coordinate system in use as the coordinate information is solely required for spatially coupling the grids.

As expressed before, GLOFRIM employs the BMI's functionalities to couple hydrological to hydrodynamic processes. Even

though the current version of GLOFRIM only supports one-directional coupling, basing it upon the BMI yields strong advantages for future two-directional coupling as coupled models do not get unnecessarily entangled. Eventually, only certain arrays of, for example, inundation depth obtained in the hydrodynamic model needs to be linked with actual evaporation rates as well as groundwater infiltration. Such two-directional coupling is currently not yet available for GLOFRIM due to on-going testing as well as concept development and will be provided in a future version of the

framework.

Besides being openly accessible and thus adaptable as well as extendable to the user's preferences or individual modelling requirements, GLOFRIM contains a number of additional advantages: first, by having PCR-GLOBWB, or any other GHM, as the hydrological output creator, the framework can easily be applied anywhere on the globe given a hydrodynamic schematization; second, models to be coupled may be selected depending on their local performance, thus possibly capturing

more relevant processes; third, the spatially explicit coupling scheme can be extended to a full feedback-loop between hydrology and hydrodynamic, also incorporating important groundwater infiltration and evaporation processes; fourth, by guaranteeing identical hydrological forcing, applying the computational framework facilitates benchmarking of hydrodynamic models by eliminating a sources of difference, potentially supporting hydrodynamic ensemble modelling approaches.

## 4 The Synthetic Test Cases

### 4.1 Set-up

To gain insight in possible differences in model behaviour between LFP and DFM, we created two synthetic test cases, one being set-up as 1-D only and the other as 2-D only. For the latter, both models were schematized such that they cover a domain of 11 cells by 500 cells, with the cell resolution being 1 km. For the 1-D only design, the channel had a length of 500

cells with a 1 km resolution, a uniform channel width of 500 m, and a uniform channel depth of 3 m. As default settings, we applied Manning's surface roughness coefficients of 0.04 s $m^{-1/3}$ for the 1-D run and 0.07 s $m^{-1/3}$ for the 2-D run. Both synthetic test cases were forced with an artificial upstream discharge boundary spanning one year and consisting of two peak flow moments to introduce variability in model dynamics. As a downstream boundary condition a constant water level of 0




m was set. The entire simulation period was three years to ensure that all water has drained before the end of the run. To assess model output, seven cross-sections were defined, hence capturing the downstream propagation of the artificial flood waves and facilitating the assessment of possible attenuation and dampening effects. For benchmarking the models we then compared discharge along the cross-sections as well as run times to obtain a first indication how the different computational

schemes might vary (Figure 2).

### 4.2 Results and Discussion

Assessing the results for both 2-D and 1-D, we find that both models simulate the same responses to the input signal applied (Figure 3). Due to the higher friction coefficient and the wider flow area, it takes the 2-D schematization almost the entire simulation period to entirely convey the water volumes to the downstream boundary. In the 1-D schematization, however, all

water is already drained after around 30 per cent of the entire simulation period. The similarity of simulated discharge between LFP and DFM is, despite the models' differences in complexity and design, in line with the findings made by Neal et al. (2012a) and De Almeida and Bates (2013). In the latter study, differences in governing equations were assessed analytically for various flow regimes ranging from sub- to supercritical flow. It was concluded that for applications with low Froude numbers ($Fr \ll 0.5$), such as the synthetic test case used here, no significant differences occur between models

solving the LIEs and those solving the full dynamics of the SWEs. Also Neal et al. (2012a) showed that it appears unnecessary to employ models solving the SWEs for flow gradually varying in time and for subcritical flow regimes. In addition, the study showed that for those applications, run times of local inertia models are shorter than those of models solving the full SWEs. The run times measured for the various synthetic test cases used here underpin this finding as LFP exhibits shorter run times, in particular for the 2-D schematization (

Table 1). To facilitate comparability, we a priori set the maximum solver time step in DFM to the average of the time steps required by LFP. It is noteworthy that the differences in run times may not merely be attributable to varying solver complexity, but partially also to the programming language and compiler used as well as to general model complexity and level of code optimization applied.

### 5 Test case: the Amazon River basin

### 5.1 Set-up

To test GLOFRIM in an actual test case as well as to benchmark the flexible and regular grid, the framework was applied in the Amazon River basin with DFM and LFP being schematized as a flexible mesh and regular grid, respectively. The methods applied to derive the hydrodynamic schematization of the Amazon River basin for DFM are explained in detail in Hoch et al. (2017). First, a regular 2-D grid at 10 km x 10 km resolution refined until a grid size of 2 km x 2 km was locally

obtained, based on the Height Above Nearest Drainage algorithm (HAND; Rennó et al. (2008)). Thereby areas with low HAND values were stronger refined than those with higher values, resulting in a finer mesh along and next to river channels. This implies a major difference to the synthetic test case above, as we now employ a flexible mesh instead of a regular grid for DFM. As input elevation, canopy-free elevation data at 15 arcsec spatial resolution was applied (Baugh et al., 2013; O'Loughlin et al., 2016) and subsequently smoothed to eliminate local depressions and other residues due to vertical errors

of SRTM data (Yamazaki et al., 2012). Elevation data was then assigned to the flexible mesh by spatial averaging. For the 1-D channel network and bathymetry, global river width data (Yamazaki et al., 2014) was employed which was combined with



the equations from Paiva et al. (2011) to derive bathymetry information. For further information, we refer to the relevant papers.

To obtain a LPF schematization equivalent to the DFM schematization, elevation data as well as both river width and river depth information were processed to agree with the requirements of LFP. For river channel properties, the depth and width

information stored in the vector data used for DFM were rasterized, and for the elevation data the smoothed canopy-free elevation data was upscaled to a 2 km spatial resolution which equals the finest spatial resolution of the DFM schematization (Figure 4). From Figure 4 it is visible that LFP contains a greater level of detail in areas farther upstream due to the finer spatial resolution uniformly applied. As a consequence, the total number of cells in LFP exceeds the number of 2-D cells in DFM by a factor 4 (Table 2). Furthermore, only around 10 per cent of the entire schematization represents 1-D channels in

LFP, while the channel network of DFM was based on around 30 per cent of all DFM cells. For both DFM and LFP, Manning's surface roughness coefficient was uniformly set to 0.03 s m$^{-1/3}$ for channel and floodplains which is consistent with other case studies in the Amazon (Paiva et al., 2013; Rudorff et al., 2014a, 2014b; Trigg et al., 2009; Yamazaki et al., 2011).

For the hydrological model PCR-GLOBWB, the kinematic wave approach was used for routing outside of the coupled

domain. This is required as the hydrodynamic schematizations do not cover the entire extent of the Amazon River basin. Since simulated discharge from PCR for the Amazon substantially under-predicts observations, we decided to apply a regionalized optimization technique facilitating comparison between simulated and measured discharge value (Hoch et al. (2017)). In analogy to the hydrodynamic models, the surface roughness coefficient of PCR was uniformly set to 0.03 s m$^{-1/3}$.

Model output of both set-ups was validated against observed GRDC-discharge at Óbidos, the most downstream station of the

GRDC-network in the Amazon River basin (Figure 4). To that end, Pearson's r, the relative mean square error RMSE, and the Kling-Gupta Efficiency KGE (Gupta et al., 2009) were computed. The model time covers the period from 01/1984 until 12/1990 with the first year being used for spin-up of the coupled setting. This period had to be chosen due to the limitation of available GRDC data for model validation. As with the synthetic test case, run times were compared. To be able to understand water level dynamics as simulated by both models, we compared them at three locations throughout the basin

(Figure 4). The locations were chosen such that they represent the upstream (Loc1), midstream (Loc2), and downstream dynamics in the basin (Loc3). Besides, inundation extent was benchmarked by applying three evaluation functions, using the LFP inundation results as the benchmark dataset. First, the hit rate H was computed based on the subsequent equation:

$$H = \frac{N_{DFM} \cap N_{LFP}}{N_{LFP}} \qquad (6)$$

$N_{LFP}$ and $N_{DFM}$ indicate thereby the number of inundated cells in LFP and DFM at the same moment in time, respectively. To

perform consistent benchmarking, the flexible cells of DFM were resembled to the resolution of LFP. The hit rate can vary between 0, signalling that DFM and LFP have no inundated cells in common and 1, indicating that all cells in LFP are also inundated by DFM.

In addition we determined the false alarm ratio F to also take into account false positive alarms. The false alarm ratio can be obtained with


$$F = \frac{N_{DFM} \setminus N_{LFP}}{N_{DFM} \cap N_{LFP} + N_{DFM} \setminus N_{LFP}} \qquad (7)$$

In the optimal situation, F would be 0 showing that no cells are incorrectly marked as flooded in DFM, whereas a value of 1 indicates that all cells are classified as false alarms.





Last, we assessed the critical success index C which combines both hit rate and false alarm ratio into one parameter which can vary between 0 in the worst and 1 in the best scenario, indicating perfect match between both inundation maps:

$$C = \frac{N_{DFM} \cap N_{LFP}}{N_{DFM} \cup N_{LFP}} \tag{8}$$

For both set-ups, the River-Floodplain-Scheme was activated and flux-updating was opted for. All simulations were

performed on a Linux environment with an Intel i7-4790 core at 3.90 GHz and 16 GB memory.

## 5.2 Results and Discussion

Benchmarking discharge results against observation from GRDC at Óbidos shows that both models behave similarly. However, LFP tends to compute earlier peak flow as well as earlier and lower low flow (Figure 5). As a consequence, obtained coefficients of correlation are lower for LFP, while the model's skill as expressed by KGE are higher for LFP and

the RMSEs are comparable (Table 3). Even though the discrepancies in simulated discharge between the two models are not remarkable, they require further investigation as they cannot be exhaustively explained with our current process understanding. Based on the results obtained in the synthetic test case and since the hydrological forcing of both models is equal in terms of water volumes, spatial distribution, and timing, we decided to evaluate the impact of the following parameters: the actual river length and dimension in LFP compared to DFM and the sensitivity of LFP to Manning's surface

roughness coefficient over large areas.

Since the routing scheme of LFP is based on a D4 system where water can flow in southerly, northerly, easterly or westerly direction, channel length and dimension in LPF tend to be longer than in other hydrodynamic models that are not based on such a system, for example DFM. Reducing the unitless meandering coefficient in LFP to scale river length, however, did not show any significant impact on simulated discharge. After investigating how changes in surface roughness values may

affect discharge estimates from LFP, we indeed found different responses to variations in surface roughness than DFM. Yet, we know from the synthetic example that both models can produce similar results when using the same friction coefficient and since the flow regime in the Amazon basin can be described as sub-critical, different sensitivity to surface roughness over large areas can be disregarded as cause for discharge discrepancies. For the remaining gap in simulated discharge, we can at this point only make assumptions about the cause. Possible reasons include differences in internal processing of 1-D

channel bathymetry, channel-floodplain interaction, and input elevation assignment due to the different gridding approaches applied.

Assessing differences in simulated water level dynamics at the observation locations, we cannot find any particularly prevailing difference between the models' response to hydrological forcing (Figure 6). In general we observe that modelled water levels are comparable, yet with locally differing patterns. While at the most upstream station Loc3 DFM simulates

lower water levels than LFP, this is opposite at the most downstream station Loc1, and at Loc2 both models provide comparable results. Besides differences in actual water levels, both models show a comparable response to model input, yet LFP tends to yield earlier peak water levels than DFM which concurs with the discharge dynamics observable. The reason for differences in simulated water levels as well as their dynamics could not be fully attributed to one specific cause. For example, the more pronounced difference in water levels at Loc1 may simply be a local effect and may be related to slight

differences in model schematization at the downstream boundary or to backwater effects in the delta regions affecting results differently. Furthermore, discrepancies are likely to be related to differences in surface elevation simulated at the observation stations due to the differences in gridding between DFM and LFP. Assessing the local properties of the observation stations revealed that the surface elevation in DFM is higher than in LFP, and due to the flexible meshing, cell size can vary greatly





too (Figure 4). Differences in cell size and therefore gridding may thus also have locally impacted the overall water levels as well.

Regarding the run times of the two coupled set-ups, we find that it takes LFP around six hours to simulate the entire simulation period of seven years, that is model time plus spin-up, while performing the same simulation with DFM takes
around seven hours (Table 3). The difference in run times is less pronounced than for the synthetic test case, which can be related to the lower number of cells in DFM compared to LFP due to use of a flexible mesh. In addition, a more computationally expensive interaction between 1-D and 2-D domain in DFM could also affect run times. As DFM is in general a multi-purpose tool whose application is not limited to inundation modelling, it is not unexpected that it may be slightly slower than programmes specifically tailored for efficient large-scale inundation modelling such as LFP.

We find that inundation extents obtained at the end of the simulation runs with DFM and LFP are comparable, yet far from identical (Figure 7). Due to the larger inundation extent of DFM, a hit rate of 0.85 is obtained, indicating that 85 % of extent as simulated by LFP is also simulated by DFM. Especially differences in inundated extent in upstream areas and along small reaches can explain the obtained false alarm ratio of 0.50 (

Table 5). These differences are also responsible for the critical success index of 0.46 corroborating that in bit less than half
of the cells inundation extent is simulated by both models. A model agreement of 46 % is slightly higher than the 30%-40% found by Trigg et al. (2016) for a benchmarking study of global flood hazard models. This in fact suggests that the choice of numerical scheme and model schematization alone can greatly impact upon inundation, confirming that differences in model forcing and boundary conditions do not act alone as a cause of modelled inundation difference, which could have been the case in the results obtained by Trigg et al. (2016) .

A main cause for the differences observed for regions further upstream is that DFM tends to compute larger flood extent than LFP: with DFM having larger cells in upstream areas due to the flexible meshing, a larger 2-D area is instantly marked as inundated for DFM once overbank flow occurs. This loss of level of detail in DFM is the concession to be made for a reduced number of grid cells and hence faster computations in the 2-D domain. For more downstream regions, differences in inundation extent are primarily present at small river channels while floodplain inundation is comparable. This, however, can
to some extent be attributed to differences in how the 1-D domain is implemented in the models, with DFM using grid-size independent vectors and LFP using grids at the overall spatial resolution of the schematization. Given the overall larger inundation extent simulated by DFM, the above-discussed deviations in simulated discharge and in particular the more pronounced wave attenuation in DFM may be explained as return flows from the floodplain to the channel seem to be faster in LFP than in DFM.

## 6 Conclusion and recommendations

In this study, we presented GLOFRIM, a GLObally applicable computational FRamework for Integrated hydrological-hydrodynamic Modelling. In its current version, it provides an environment to one-directionally couple the global hydrological model PCR-GLOBWB (PCR) with two hydrodynamic models: Delft3D Flexible Mesh (DFM) solving the full shallow-water equations, and LISLFOOD-FP (LFP) solving the local inertia equations. By linking hydrology to
hydrodynamics, it is possible to take advantage of the strengths of both while at the same time compensating their weaknesses.

We define five main assets of GLOFRIM: (i) it is openly accessible and hence can be directly applied, adapted to specific purposes, and extended with other models; (ii) by employing a global hydrological model to obtain model forcing, the framework can easily be applied globally;  (iii) models to be coupled may be selected depending on their local performance



and thus more relevant processes can be captured; (iv) the spatially explicit coupling scheme can be extended to a full feedback-loop between hydrology and hydrodynamics; (v) thorough benchmarking and ensemble modelling of hydrodynamic models is supported by providing identical hydrological forcing for experiments.

GLOFRIM at present provides a range of possible options for model coupling. Users can choose between coupling PCR to
either the 1-D or 2-D domain, can specify whether to update hydrodynamics through states or fluxes, and can run hydrodynamic models in both spherical and projected coordinate systems. It is generically written and does not require any a priori knowledge of the code as all important settings are specified in a separate settings-file.

Besides PCR as well as DFM and LFP, there are a number of other global hydrological and hydrodynamic models available which have their individual advantages. As the framework is freely and openly available, its design can easily be extended
and adapted to cater the coupling of other hydrological or hydrodynamic models, merely requiring the implementation of the BMI into each model to be added. The BMI does not change the model functionality while at the same time providing a range of added functions. Furthermore, not all model variables need to be exposed, only those to reproduce model geometry, distinguish between 1-D and 2-D cells, and a variable to be updated. We therefore recommend considering this option for future model developments and will also aim to incorporate other models ourselves. To our knowledge, spatially explicit
model coupling at global scale by means of such a framework is unprecedented. Consequently, user experiences and lessons learnt are still sparse and any initiatives regarding framework extension are therefore kindly received by the authors, as well as feedback and experiences made. We also recommend the testing and application of it in other study areas and under different boundary conditions to further evaluate the code, process flow, and applicability.

Before applying GLOFRIM in an actual test case, we performed a simple synthetic test case to obtain a first-order insight in
how both models may differ regarding their computational complexity. Thereby both the 1-D and 2-D domain were forced by a synthetic inflow signal and simulated discharge was evaluated along the flow path. Results show that both models produce the same response to the signal despite the difference in solver complexity. The results obtained are in line with previous studies showing that for sub-critical flow regimes discharge results should be similar (De Almeida and Bates, 2013; Neal et al., 2012a).

Both hydrodynamic models were then applied within GLOFRIM and evaluated regarding simulated discharge, water levels, run time, and inundation extent, also constituting a first comparison of large-scale flexible mesh and regular grid applications. Assessing simulated discharge for the test case in the Amazon River basin shows that both models exhibit comparable results with LFP tending to compute earlier and slightly increased peak discharge estimates. As thorough testing of possible causes did not show significant improvements, we speculate that differences in processing of 1-D channel
bathymetry, interaction between 1-D channels and 2-D floodplains or assignment of surface elevation data to the different grids may impact discharge results. A more in-depth analysis of these differences was however outside the scope of this study and thus needs to be performed in a follow-up study. As the general overprediction of observed discharge at Óbidos can partly be attributed to the absence of hydrological processes on inundated floodplains, it is envisaged to extend the current code such that it also caters for a full feedback loop between hydrodynamics and hydrology.

Water levels simulated by both models differ locally, yet only slightly. These discrepancies between both models are most likely due different grid schematizations in DFM and LFP, which results in locally differing elevation values and cell areas and thus influences simulated water levels. Due to differences in model structure, downstream boundary conditions had to be implemented slightly differently, possibly also impacting water level results in particular for more downstream stations. As it was the aim of this paper to introduce the computational framework applied, a more elaborated evaluation of causes for
water level deviations is future work.



A key parameter for large-scale modelling is run time. In the current study, the schematization of LFP contains more than four times the number of 2-D cells than DFM while the number of 1-D cells is 40 per cent higher in LFP as in DFM. Despite the greater number of cells, LFP has a slightly shorter run time. This is in line with the results obtained in the synthetic test case, yet the relative difference is reduced due to the application of flexible meshes for the 2-D domain and the nature of the

coupling algorithm applied: because water was coupled directly into the 1-D channels, flow over the 2-D domain was limited and, as a result, so was the impact of differences in computational efficiency of the models. Differences in run times may also be related to more fundamental factors, such as the degree of code optimization applied. Additionally, DFM was, in contrast to LFP, not explicitly developed for efficient inundation modelling, but as a multi-purpose tool including a number of additional physical processes, such as the potential to simulate 3-D flow, estuarine processes or hydrogeomorphologic

dynamics, which could also result in longer run times. To better understand causes of run time discrepancies, further model development, testing, and evaluation is therefore recommended.

To benchmark LFP and DFM in terms of simulated inundation extent in the Amazon River basin, the hit rate H, the false alarm ratio F, and the critical success index C were determined. In general, both models agree about as often as they disagree C=0.46 indicating that both DFM and LFP predict simulation extent for around half of all cells. This level of agreement is

slightly higher than the one obtained by Trigg et al., (2016) and is a strong indication that the model geometry and numerical scheme play a similarly strong role in influencing model accuracy as the boundary conditions and model forcing applied in global flood hazard models. Moreover, a higher value could not be obtained due to the impact of the flexible mesh, especially for upstream areas where DFM runs at cells that are a factor 25 larger than in LFP. While such large cells contribute strongly to shorter run times, they may also have implications for detailed flood hazard estimates which can be

strongly hampered. In case of employing a flexible mesh it seems as if an a priori decision has to be made where and to which extent such models are supposed to provide fine-scale results or whether computational efficiency is the main aim – both at the same time does not seem to be feasible from our results. We hence recommend testing the application of flexible meshes for riverine inundation modelling in more detail to obtain a better understanding of the trade-off to be made between grid refinement and related run time. Besides, further benchmarking of the impact of flexible meshes on model accuracy

with respect to regular grids is recommended.

With the presented computational framework GLOFRIM and the satisfactory results obtained, we trust to have contributed to the current development of model coupling and integration, and to have provided an openly accessible tool that facilitates more accurate large-scale flood hazard estimates. We hope that, eventually, the integration of hydrological and hydrodynamic models will lead to improved flood risk assessments and planning of climate change impact mitigation and

adaption measures.

*Code and/or data availability.* The code of GLOFRIM as well as the BMI-versions of LISFLOOD-FP and PCR-GLOBWB are openly accessible and freely downloadable at doi.org/10.5281/zenodo.597107

*Author contributions.* Fedor Baart and Jannis M. Hoch developed the BMI adapter for LISFLOOD-FP. Jannis M. Hoch, Rens van Beek, and Jeffrey C. Neal developed the code of the computational framework. Jeffrey C. Neal, Rens van Beek, Hessel C. Winsemius, Paul D. Bates, and Marc F. P. Bierkens supervised the research and provided important advice. Jannis M. Hoch designed and executed the research, and also prepared the manuscript as well as code, with contribution from all co-authors.


*Competing interests.* Author Jeffrey C. Neal is a topical editor of the journal.



*Acknowledgements.* The authors declare that they have no conflict of interest. This study was financed by the EIT Climate-KIC programme under project title "Global high-resolution database of current and future river flood hazard to support planning, adaption and re-insurance". We also want to acknowledge the contributions of Climate-KIC and University of

5   Bristol to realize a research stay at the University of Bristol. Special thanks are reserved for Arthur van Dam and Herman Kernkamp from Deltares for their support in applying Delft3D Flexible Mesh.



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



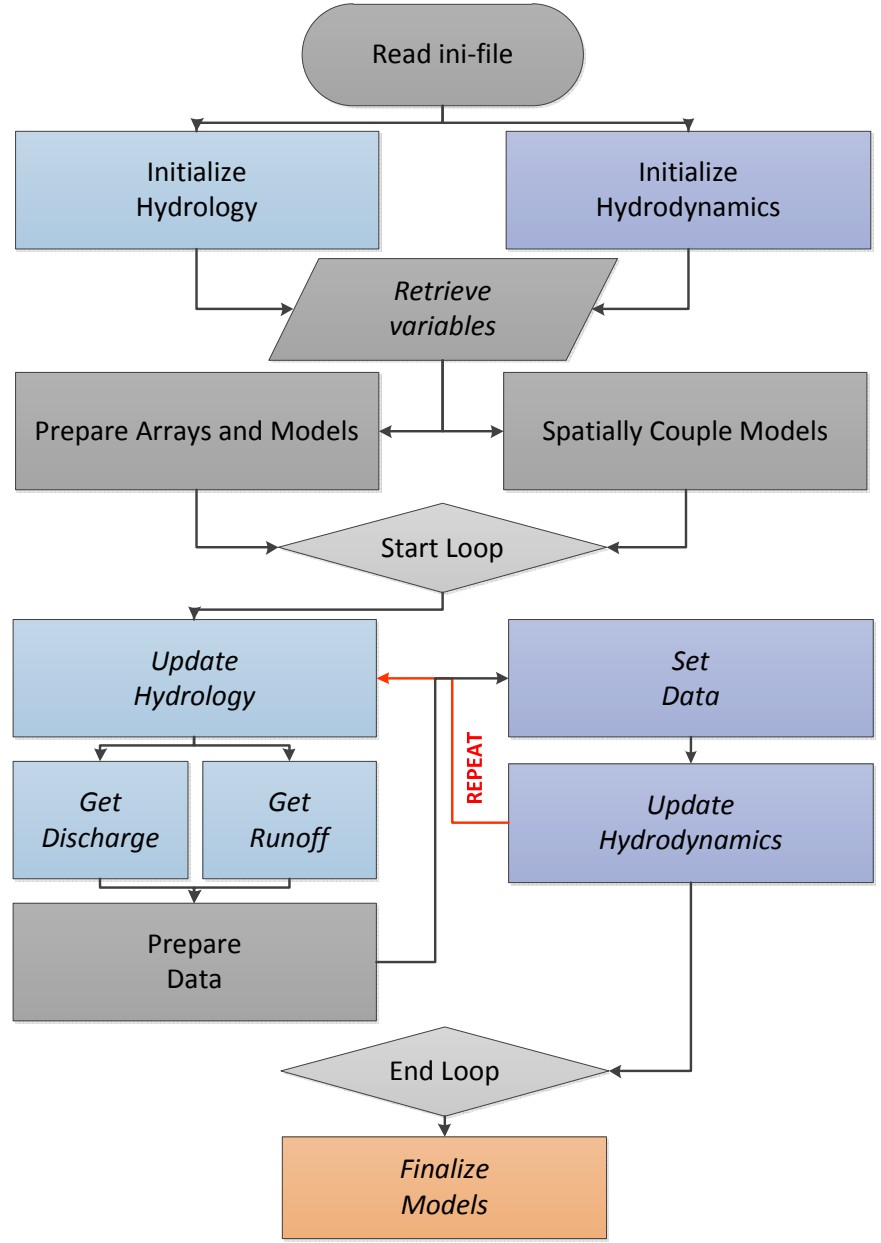

**Figure 1: Flow diagram of steps executed in computational framework; all steps in italic are taken by using the Basic Model Interface (BMI)**



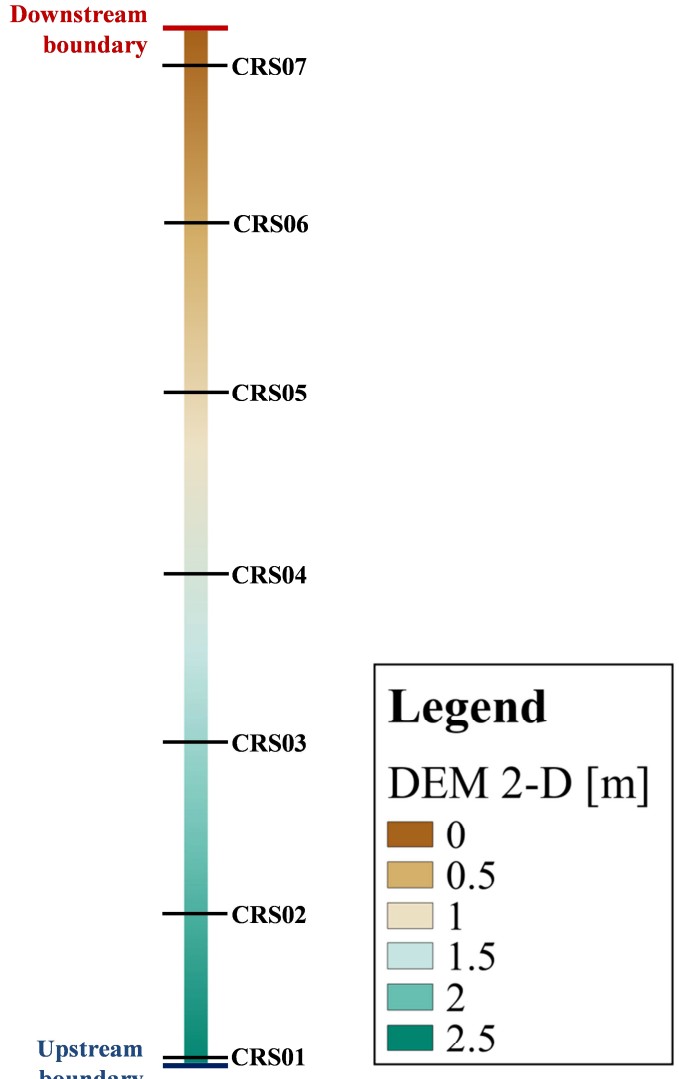

Figure 2: DEM of the 2-D synthetic test case for LFP and DFM.



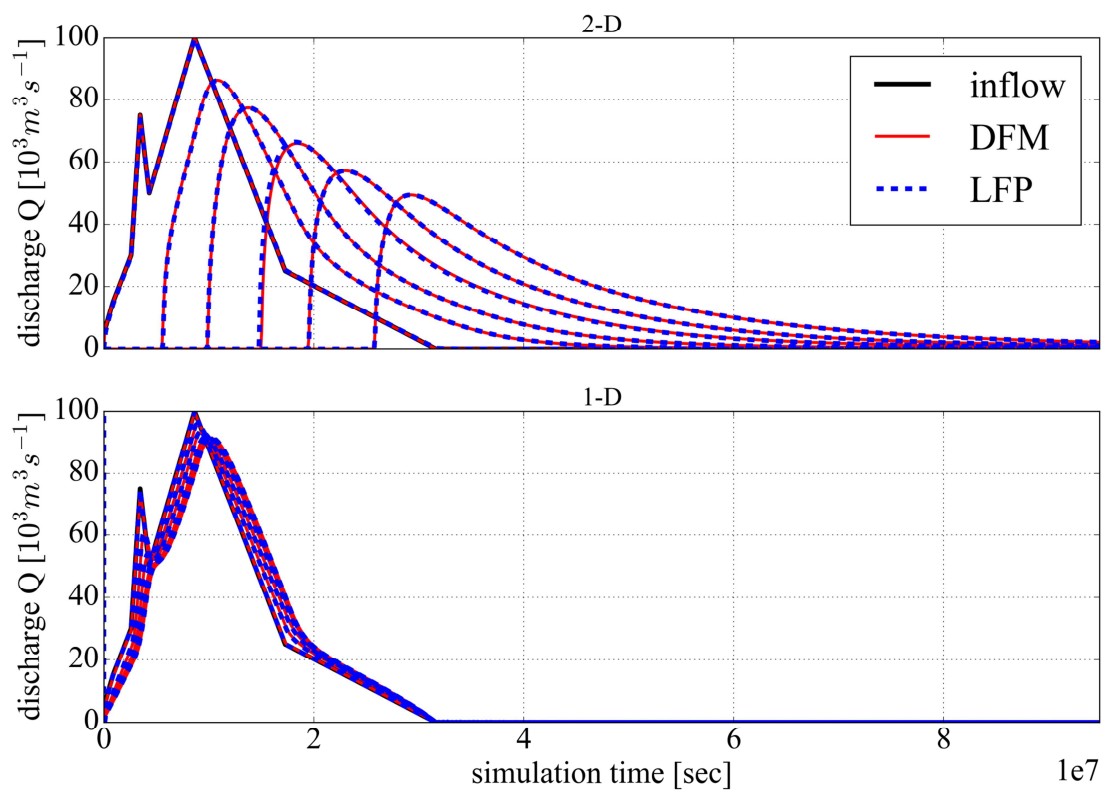

**Figure 3: Simulated discharge of both 2-D and 1-D synthetic test case; the inflow curve is not well visible as it virtually coincides with observed discharge at CRS01**

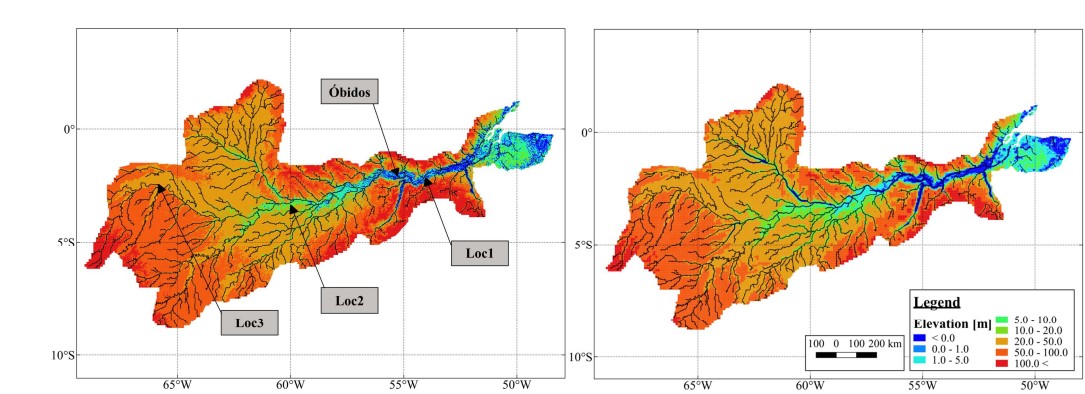

**Figure 4: Digital Elevation Model as well as 1-D channel network as used in LISFLOOD-FP (left) and Delft3D Flexible Mesh (right); discharge was benchmarked and validated at Óbidos while water levels were compared at three locations throughout the domain**



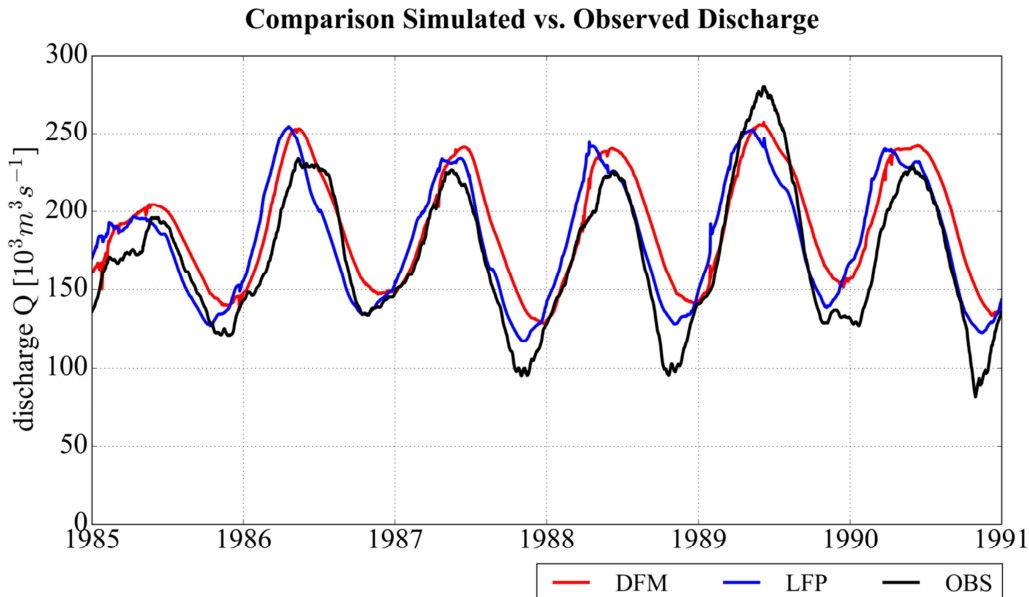

**Figure 5: Observed discharge from the Global Discharge Data Centre (GRDC) as well as simulated discharge from both DFM and LFP at Óbidos**

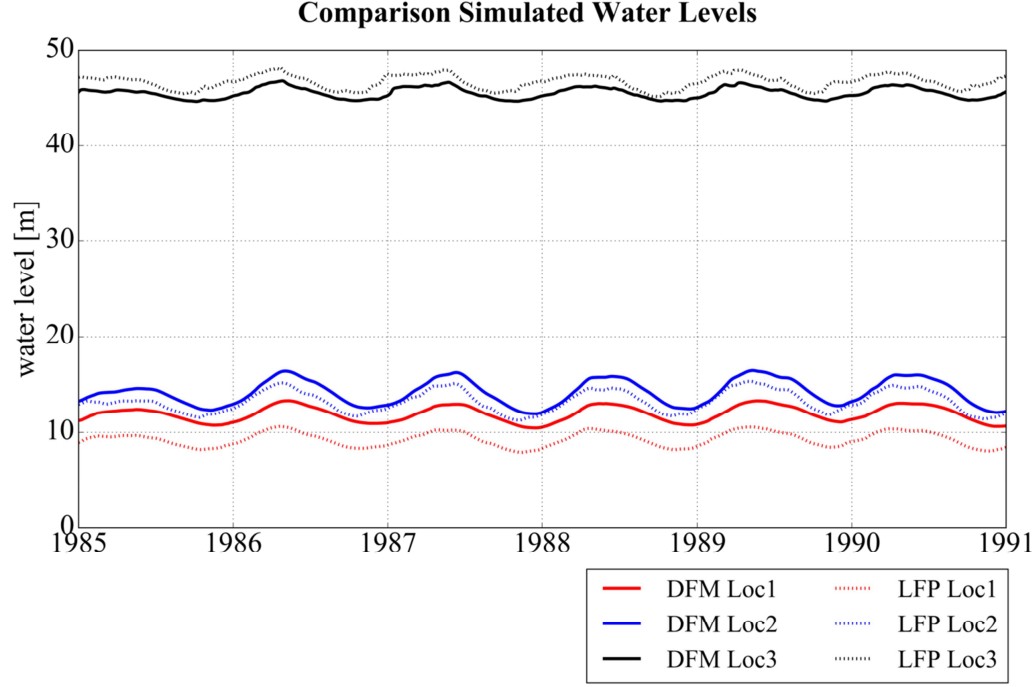

5    **Figure 6: Comparison of simulated water depth at three different locations randomly picked within the domain**



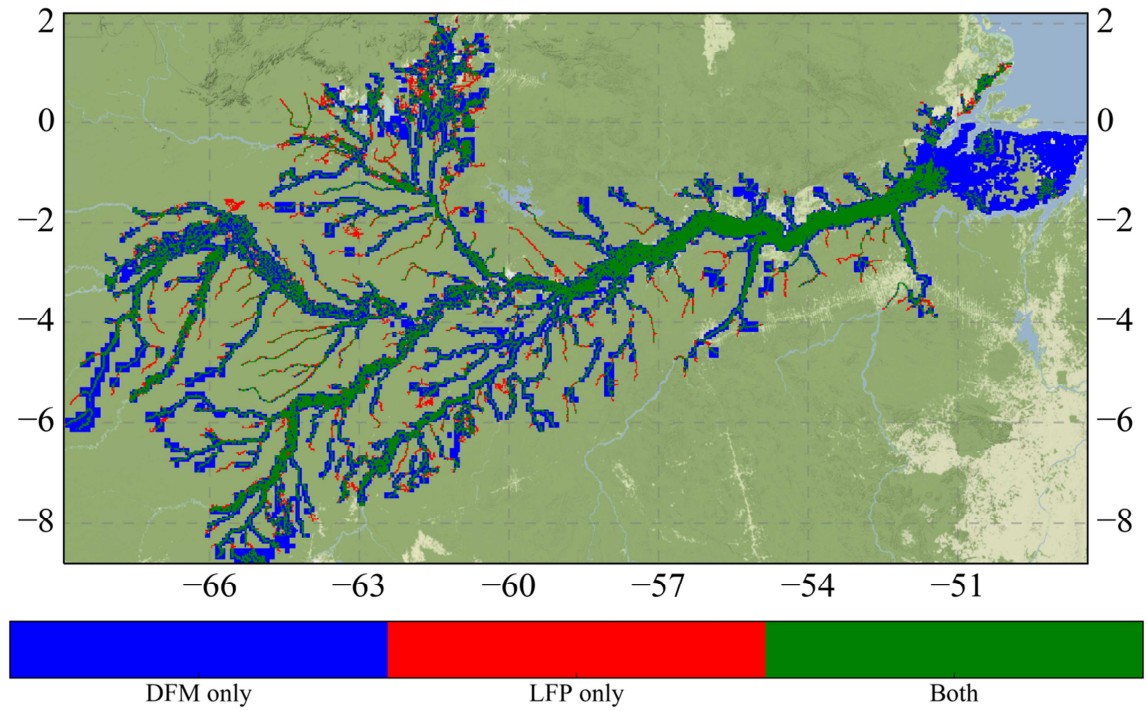

**Figure 7: Benchmarking simulated inundation extent by DFM and LFP.**

**Table 1: Run times of different set-ups in synthetic test case**

|  | **2-D** | **1-D** |
|---|---|---|
| **DFM** | 19.5 min | 5.5 min |
| **LFP** | 2.1 min | 2.6 min |

5  **Table 2: Overview of key properties of hydrodynamic schematizations coupled to PCR-GLOBWB in this study**

|  | **2-D cells** | **1-D cells** | **Smallest cell size** | **Largest cell size** |
|---|---|---|---|---|
| **DFM** | 41,207 | 12,185 | 2 x 2 km | 10 x 10 km |
| **LFP** | 174,982 | 17,119 | 2 x 2 km | 2 x 2 km |




**Table 3: Results of Pearson's coefficient r, root mean square error RMSE, and Kling-Gupta-Efficiency KGE obtained to benchmark discharge as well as run times of coupled runs**

|  | r | RMSE | KGE | Run time |
|---|---|---|---|---|
| DFM | 0.92 | 25,289 m$^3$ | 0.76 | 7 h |
| LFP | 0.89 | 22,291 m$^3$ | 0.82 | 6 h |

**Table 4: Local properties of water level observation stations; input elevation refers to values obtained after hydraulic conditioning of canopy-free SRTM elevation data at 15 arcsec spatial resolution**

|  | Loc1 | Loc2 | Loc3 |
|---|---|---|---|
| Input elevation | 4.0 | 7.0 | 44.5 |
| Model elevation LFP | -0.2 | 2.4 | 37.4 |
| Model elevation DFM | 0.5 | 4.9 | 42.5 |
| Cell area LFP | ~4 x 10$^6$ | | |
| Cell area DFM | 7,7 x 10$^6$ | 7,7 x 10$^6$ | 30,9 x 10$^6$ |

**Table 5: Resulting benchmarking indicators for inundation extent**

|  | H | F | C |
|---|---|---|---|
| LFP / DFM | 0.85 | 0.50 | 0.46 |

