# Peer review of "GLOFRIM v1.0 – A globally applicable computational framework for integrated hydrological-hydrodynamic modelling"

_Geoscientific Model Development, 2017_

## Referee Comment (RC1) · Anonymous Referee #1 · 8 Aug 2017

This manuscript introduces GLOFRIM, a framework for coupling hydrological and hydrodynamic models. The authors test this framework using the PCR-GLOBWB model and two hydrodynamic models. Overall, I found this manuscript very well written and concise and feel it could make a significant contribution to its field. However, I do have some minor comments that I would like addressed:

Page 2, Line 5: 'Sound inundation estimates'. Could the authors use a less colloquial term. Page 4, Line 36: Please provide the spatial resolution of the CRU data. Section 2: A schematic of the models used would be beneficial to the reader. Page 7, Line 37: Provide reference for the SRTM data. Page 11, Line 7: Did you take into account

uncertainties in the discharge at Obidos? Page 11, Lines 14-20: Why are the results of the sensitivity analysis not included? The results are not surprising but you need to provide evidence. Page 11, Line 25-26: What are the different gridding approaches applied? I'm not sure if this is stated elsewhere in the manuscript.

―――――――――――――――――――

---

## Referee Comment (RC2) · D. Yamazaki (Referee) · 14 Aug 2017

<General Comments>

This manuscript introduces a new framework for a coupled "land hydrology & river hydrodynamics" modelling, and assessed its feasibility by comparing the results of two river hydrodynamic models (i.e. Delft3D Flexible Mesh and LISFLOOD-FP). The manuscript is well written, and it suits with the subject of the journal "Geoscientific Model Development". However, it contains some unclear points which should be revised/improved before publication.

[Figure]

<Specific Comments>

P7. L32: "If RFS is activated, water volume is directly coupled to the 1-D channels of the hydrodynamic model while, when RFS is inactive, water is distributed over all grid cells of the 2-D domain."

The description of RFS is not sufficient. Please explain the relationship between the hydrology model grid and hydrodynamic model pixels in a more detailed manner. I guess, "water volume of each coarse-resolution hydrology model grid" is distributed to the "corresponding high-resolution hydrodynamic model cells within the coarse-resolution grid". And the difference due to RFS is whether water volume is distributed only to river cells or both river and floodplain cells within each coarse-resolution grid box. Readers who are not familiar with this topic might misunderstood water volume is distributed uniformly all-over the calculation domain (not to the corresponding cells).

P7. L35: "the accuracy of the 2-D elevation data which is known to contain strong vertical bias, in particular when derived from remotely sensed global data"

The elevation data is affected not only by vertical bias but also by various random/systematic noises. I recommend to add reference to the latest research on this topic [Yamazaki et al., 2017]

P8. L4: "We found that updating fluxes reduces run times compared to states, and hence advise opting for for this option."

How downstream boundary conditions are treated. The two test cases executed in the manuscript assumes the downstream boundary is river mouth (0m constant). Some potential users might be interested to simulate flooding in middle-stream, that requires a setting of downstream boundary conditions. Without a reasonable treatment of the downstream boundaries, it is difficult to state that the developed framework is "globally applicable". [I also note that "for" appears twice in the sentence.]

P8. L9: "although PCR runs in spherical coordinates."

Given that a spherical coordination can be organized by a Cartesian system, it's better to clarify that PCR-GLOBWB runs at "non-Cartesian spherical system" while it is possible to use "regular lon-lat Cartesian system" for hydrodynamic models.

P8. L30. "4 The Synthetic Test Cases"

It's better to explicitly state that PCR-GLOBWB is not used in the synthetic test case. I think this test case is done only for comparing Delft3D and LISFLOOD-FP under an ideal situation. Thus, this test case is directly not related to the GLOFRIM framework, thus readers might be confused.

P8. L34: "0.04 s m-1/3 for the 1-D run and 0.07 s m-1/3 for the 2-D run."

Does this mean the same roughness coefficient is used for river channel and floodplains in 2D run? Usually, river channels have smaller roughness compared to floodplains.

P9. L25: "5.1 Set-up"

Please describe how the downstream boundary was treated as this is critical for simulations. Please also explain how the complex channel network of the delta, bifurcating sections, and braided streams were treated. If they are treated differently by Delft3D and LISFLOOD-FP, this difference could be a potential cause of the disagreement of the simulation results.

P10. L6. "for the elevation data the smoothed 5 canopy-free elevation data was upscaled to a 2 km spatial resolution"

Please explicitly explain how the DEM was upscaled, because this has large impact on flood inundation. Did the authors took the mean within a cell, or the minimum elevation?

P10. L11. "roughness coefficient was uniformly set to 0.03 s m-1/3 for channel and floodplains".

Was the same roughness used for channel and floodplain? If so, please clearly state,

Interactive
comment

because different values are usually used.

P10. L14: "For the hydrological model PCR-GLOBWB, the kinematic wave approach was used for routing outside of the coupled domain."

Please explain that the simple kinematic routing may result in poor upstream boundary inflow, as backwater effects or river floodplain interactions can be neglected. Also, this approach could be a limitation for generating a realistic downstream boundary condition. Probably, using continental-scale hydrodynamic model (such as MGB-IPH or CaMa-Flood) as an intermediate step between the hydrology model and high-resolution hydrodynamic model can be a solution.

P10. L16: "we decided to apply a regionalized optimization technique"

This "regionalized optimization" could be a limitation for using the proposed framework for "global application". The "modelling of flood inundation" may be possible at a global scale, but "global application" can be restricted by the quality of input/boundary datasets. Please state this limitation in the conclusion section.

P11. L6: "5.2 Results and Discussion"

More detailed analysis of the difference between Delft3D and LISFLOOD-FP is needed. As far as I guess from the figures, the flood peak of Delft3D is later than LISFLOOD-FP because it has larger inundation in upstream areas due to its coarser flexible cell resolution. The smaller water level amplitude in upstream must be also related to the larger inundation in upstream. Because floodplain inundation attenuate flood waves, suppress water level fluctuations, and delays flood peak, most of the disagreement between the two models can be explained consistently due to the inundation in upstream regions. The authors can analyze this effect easily, by comparing the simulated discharges by Delft3D and LISFLOOD-FP also at upstream locations other than the Obidos. By analyzing discharge, we can show where flood waves were attenuated. I suggest to include this discussion in the manuscript. And if the discussion

above is true, the Delft3D simulation must be sensitive also to the spatial resolution in upstream regions, thus I recommend to include some sensitivity test on the spatial resolutions.

P11. L16: "Since the routing scheme of LFP is based on a D4 system, channel length and dimension in LPF tend to be longer than in other hydrodynamic models"

This is not precise. The D4 river network can generate shorter channel length if the scale of channel meandering is smaller than the size of the cell. Please rewrite this sentence. Furthermore, the D4 system does not only change the flow length. It alters the connectivity of channels and floodplains. Some channels/floodplains which are connected in the D8 system (or a vector system) could be disconnected in the D4 system, because diagonal connectivity is not allowed. To avoid this problem, the DEM should be adjusted to ensure the D4 connectivity. Please make a discussion on this issue, as this could be one of the main reason of the difference between Delft3D and LISFLOOD-FP.

P11. L29: "While at the most upstream station Loc3 DFM simulates lower water levels than LFP"

This is probably due to larger flooding in Delft3D due to its coarser spatial resolution in upstream, as discussed above. Please clarify.

P11. L34: "the more pronounced difference in water levels at Loc1 may simply be a local effect"

What is the "local effect". Please explain in detail.

<References> Yamazaki et al. (2017), A high accuracy map of global terrain elevations, Geophysical Research Letters, doi: 10.1002/2017GL072874

---

## Short Comment (SC1) · 29 Aug 2017

The Authors wrote that "hydrodynamic models lack an advanced implementation of hydrological processes". Attempts have been made in this direction, in which a full feedback loop between hydrodynamics and hydrology is considered, and deserve to be mentioned (Kim et al., 2012; Viero et al., 2014).

Flooding patterns in populated environments are crucially affected by small-scale, linear features such as artificial embankments (roads, railways, levees of minor channels, etc.). To be applied at the global scale, the spatial resolution of a hydrodynamic model is necessarily coarse with respect to the width of these linear features. How can such a

"coarse" grid account for these effects? For instance, Vacondio et al. (2016) resampled a refined DTM to a coarser grid by retaining the highest terrain elevation, but such a strategy has obvious shortcomings if applied to cell size of the order of 1 km. Certainly, the use of a flexible mesh can be of aid.

Generally speaking, I fear that thinking of flooding "at a global scale" entails the risk of neglecting features and factors that are small from, e.g., a pure geometrical point of view, but play a major role in lowlands hydrodynamics. Modellers should always be aware of such a risk.

References:

Kim, J., A. Warnock, V. Y. Ivanov, and N. D. Katopodes (2012), Coupled modeling of hydrologic and hydrodynamic processes including overland and channel flow, Adv. Water Resour., 37, 104–126, doi:10.1016/j.advwatres.2011.11.009.

Viero, D. P., P. Peruzzo, L. Carniello, and A. Defina (2014), Integrated mathematical modeling of hydrological and hydrodynamic response to rainfall events in rural lowland catchments, Water Resour. Res., 50, 5941–5957, doi:10.1002/2013WR014293.

Vacondio, R., F. Aureli, A. Ferrari, P. Mignosa, and A. Dal Palu (2016), Simulation of the January 2014 flood on the Secchia River using a fast and high-resolution 2D parallel shallow-water numerical scheme, Nat. Hazards, 80,103–125, doi:10.1007/s11069-015-1959-4.

---

## Short Comment (SC2) · 29 Aug 2017

Dear Daniele P. Viero, Thank you for your constructive remark which surely touches upon some aspects of global flood modelling worth mentioning.

I have read your literature recommendations with interest and will try to mention them where adequate in an updated version of the manuscript. There are indeed some hydraulic models that can simulate a limited number of hydrologic processes, such as groundwater infiltration, precipitation or evaporation, to a varying complexity. The focus in our statement was thus more on "advanced" as none of the current (large-scale) hydrodynamic models represent the complexity of hydrologic processes similarly well

as hydrologic models themselves.

With respect to your remark on limitations of global models, I want to draw your attention to the interesting paper of Ward et al. (2015), not only discussing these limitations, but also showing their usefulness with several examples. As much as I agree that for local flood risk management practices global models may not be the ideal choice as they cannot account for levee properties, for example, there is no reason to be in fear as they can contribute to other aspects of flood risk. For instance, global flood risk models can prove useful for hot-spot detection or operational forecasting. Obviously, clear communication of uncertainties in model results is key, but that is even so for local models.

From my point of view, global and local flood risk models should therefore not (and probably never will) be mutually exclusive, but quite the opposite. Depending on the goal of the modelling study, but also on data availability, the user should choose from either of them. It may even be possible to design a modelling chain from a first coarse-scale assessment by global models to a detailed local model that allows the involvement of local stakeholder.

References

Ward, P. J., Jongman, B., Salamon, P., Simpson, A., Bates, P. D., de Groeve, T., Muis, S., de Perez, E. C., Rudari, R., Trigg, M. A. and Winsemius, H. C.: Usefulness and limitations of global flood risk models, Nat. Clim. Chang., 5(8), 712–715, doi:10.1038/nclimate2742, 2015.

---

## Author Comment (AC1) · 31 Aug 2017

We heartily thank Dr. Dai Yamazaki for his detailed evaluation of our manuscript and for his helpful comments, which we will address point-by-point hereafter.

With respect to the first comment, it seems as if the use of jargon resulted in a lack of clarity. Thanks for pointing this out! We thus will update the manuscript such that the functionality of the River-Floodplain-Scheme (RFS) is described in a better understandable way.

Indeed, the causes for vertical inaccuracy of remotely sensed elevation data are man-

ifold. Since at the time of writing the current manuscript the MERIT DEM was not yet published, we did not refer to it. We will make up for it in the revised version.

Regarding the comment on P8/L4 we are not fully sure how the explanation and comment fit together. Considering only the comment, we thank you for mentioning the missing information. In the revised manuscript, we will add information regarding schematizations not bordering at the sea, for instance mid-stream schematizations. In short, it is possible to employ any hydrodynamic schematization within GLOFRIM as long as it complies with Delft3D Flexible Mesh and LISFLOOD-FP requirements, respectively. This means that not only mid-stream simulations can be run, but adding to your comment, it would also be possible to account for tidal variations of downstream water level boundaries, provided the required information is available.

Thank you for pointing out the lack of clarity on P8/L9 and P8/L30. We will update the manuscript accordingly.

With regard to your comment on P8/L34, we want to clarify that there is actually no channel in the 2-D run of the synthetic test case. In this run, we merely simulated floodplain flow as flow over a 2-D grid, and therefore employed only one surface roughness coefficient (0.07 s m-1/3). While the surface roughness coefficient differs compared to the 1-D run, upstream and downstream boundaries as well as gradient are identical for both runs. To avoid any confusion, we will re-formulate the section under consideration more clearly.

Your comment on P9/L25 is most helpful: both hydrodynamic models have a fixed water level of 0 m as downstream boundary condition to ensure comparability. Since the schematization of LISFLOOD-FP is derived from Delft3D Flexible Mesh, both models employ the same 1-D network. In contrast to other current global hydrodynamic models such as CaMa-Flood (Yamazaki et al., 2014b), the schematizations employed in the current manuscript cannot account for bifurcations and thus the channel complexity of the delta had to be captured with one channel. To that end, we used the Global Width

Database for Large Rivers (GWD-LR; Yamazaki et al., 2014a) to obtain an effective bathymetry, accounting for islands and other disturbances in channel network. As we acknowledge that the lack of channel representation may have influenced our models results, we will elaborate on that in the discussion part of the revised version.

Thank you for mentioning the upscaling procedure on P10/L6. In line with previously published research (Fewtrell et al., 2008; Savage et al., 2016), we applied the nearest neighbour method to avoid undesired smoothing effects in the elevation values. We will update the manuscript accordingly.

Thanks for your comment on P10/L11. We indeed used a uniform surface roughness value as this has also been done by other studies as stated in the current manuscript. Hence, we do not see the need to further elaborate on this aspect in the revised manuscript.

We thankfully acknowledge your remark on the downsides of the kinematic wave approximation (P10/L14) as you are indeed raising a relevant issue regarding the upstream inflow boundary. Depending on the flow distance covered by PCR-GLOBWB routing until the hydrodynamic domain is reached, the impact of less sophisticated routing schemes may amplify, yet a detailed analysis thereof exceeds the scope of the paper. We will, nevertheless, add this aspect to the discussion of model results. Your recommendation to use a continental-scale model, for instance CaMa-Flood or MGB-IPH, as intermediate step may be one solution to circumvent the impact of the kinematic wave approximation. As GLOFRIM does not yet allow for coupling PCR-GLOBWB with other models, this is not yet feasible but provides a great motivation for future research as the coupling framework, in principle, allows for it. A more straightforward way would be to extent the hydrodynamic schematization over the entire PCR-GLOBWB domain. However, this would result in increased run times that are impractical as yet.

You correctly pointed out that a global application can be locally restricted by data availability for model set-up and we will mention this limitation in the conclusion. We
will furthermore point out that the "regionalized optimization" (P10/L16) is optional and PCR-GLOBWB can also be run as is in its default parameterization.

We are thankful for your extended remark on the model results. First of all, we agree with the assumption made that the spatial resolution applied by Delft3D Flexible Mesh in upstream areas may impact model results locally as well as further downstream. Due to that we are currently working on a follow-up study concerning the relation of spatial resolution of flexible meshes and model results. Therefore, we desisted from providing a too elaborate discussion in the current version of the manuscript, but will nevertheless mention this interplay in the revised version. In addition, we will re-run out model set-ups with an increased number of observation points, especially for the upstream region, to obtain a clearer picture whether discharge attenuation increases along the flow path. Depending on the availability of observed data, we will perform additional discharge validation for the upstream stations as well. The discussion and conclusion sections will then be updated accordingly.

Thank you for the information on the D4 river network system and the related uncertainties on P11/L16. We will extent the description of this system and its limitations, especially with respect to model results. As the anonymous reviewer #1 already suggested to provide an additional plot with the results of the sensitivity analysis of increased/decreased river length in LISFLOOD-FP, we will put this in context with your remarks to show that possible under- or overestimation of channel dimensions due to the D4 system is not contributing to the gap compared with discharge simulated by Delft3D Flexible Mesh.

This relation between simulated water levels (P11/L29) and cell area is indeed present as suggested by the reviewer. We will clarify this in the revised manuscript.

Thank you for pointing out the lack of clarity on P11/L34. A previous study already showed that the behaviour of simulated water level is not always predictable due to spatial feedback dynamics between neighbouring cells of an observation station (Hardy

et al., 1999). We will add this vital information to the discussion of model results in the revised manuscript.

References

Fewtrell, T. J., Bates, P. D., Horritt, M. and Hunter, N. M.: Evaluating the effect of scale in flood inundation modelling in urban environments, Hydrol. Process., 22(26), 5107–5118, doi:10.1002/hyp.7148, 2008.

Hardy, R. J., Bates, P. D. and Anderson, M. G.: The importance of spatial resolution in hydraulic models for floodplain environments, J. Hydrol., 216(1), 124–136, doi:http://dx.doi.org/10.1016/S0022-1694(99)00002-5, 1999.

Savage, J. T. S., Bates, P. D., Freer, J. E., Neal, J. C. and Aronica, G. T.: When does spatial resolution become spurious in probabilistic flood inundation predictions?, Hydrol. Process., 30(13), 2014–2032, doi:10.1002/hyp.10749, 2016.

Yamazaki, D., O'Loughlin, F. E., Trigg, M. A., Miller, Z. F., Pavelsky, T. M. and Bates, P. D.: Development of the Global Width Database for Large Rivers, Water Resour. Res., 50, 2108–2123, doi:10.1002/2012WR013085.Received, 2014a.

Yamazaki, D., Sato, T., Kanae, S., Hirabayashi, Y. and Bates, P. D.: Regional flood dynamics in a bifurcating mega delta simulated in a global river model, Geophys. Res. Lett., 41(9), 3127–3135, doi:10.1002/2014GL059744, 2014b.
* * *

---

## Author Comment (AC2) · 31 Aug 2017

We thank the anonymous reviewer for his/her evaluation of our manuscript and helpful comments.

Regarding the comment on colloquial language as well as missing references or information, we will carefully revise the manuscript. Hereafter, we address the main aspects brought up by the reviewer.

First of all, the original spatial resolution of the CRU-forcing is 30 arcmin. This agrees with the spatial resolution at which PCR-GLOBWB was applied in the study. Further information concerning the CRU forcing data as well as its processing for PCR-GLOBWB can be found at http://vanbeek.geo.uu.nl/suppinfo/vanbeek2008.pdf. We will present this information and the reference more prominently in the revises version.

For model validation against discharge we did not specifically address the uncertainty of observed discharge at Obidos. We neglected this aspect as we assume that the uncertainty is insignificant compared to other possible uncertainties, for instance parameterization of PCR-GLOBWB or surface roughness of the hydrodynamic models, particularly for large-scale modelling studies. It must nevertheless be acknowledged here that the uncertainty of observation may vary between 10% and 30% due to the rating curve applied at the observation station. Clarke et al. (2000) reported an uncertainty of around 16 % of year-to-year variability. Even though an uncertainty analysis exceeds the scope of this paper, we will refer to this information in the revised manuscript. Despite their uncertainty, the available discharge observations have therefore be used as validation datasets.

With respect to the comment on the unpublished results of the sensitivity analysis, we decided to not provide an additional plot as we assumed that this may distract the reader from the core of the manuscript, that is the model framework in itself as well as the test case in the Amazon basin. Given your comment, however, we not believe that the text indeed needs to be supplemented by a figure. Therefore, an explanatory figure will be added to the revised version of the manuscript.

Last, we want to clarify which gridding approaches is referred to: flexible gridding (or "meshing" as in Delft3D Flexible Mesh) and regular gridding (as done by LISFLOOD-FP). To avoid unnecessary confusion, we will update this statement accordingly.

References

Clarke, R. T., Mendiondo, E. M. and Brusa, L. C.: Uncertainties in mean discharges from two large South American rivers due to rating curve variability, Hydrol. Sci. J., 45(2), 221–236, doi:10.1080/02626660009492321, 2000.

---

## Author Response (AR1)

**Rebuttal Letter Referee #1, Anonymous Reviewer**

We thank the anonymous referee #1 for the kind words on our manuscript and the points brought forward as they resulted in an improvement of the submitted manuscript. We have added the outcome of the sensitivity analysis of different LISFLOOD-FP parameters. Below, we repeat the reviewer's comments, and provide our response in italics. In the revised manuscript, the changes made to the manuscript are highlighted in yellow.

Page 2, Line 5: 'Sound inundation estimates'. Could the authors use a less colloquial term.

*We adjusted the manuscript accordingly and replaced too colloquial terms in general.*

Page 4, Line 36: Please provide the spatial resolution of the CRU data.

*Thank you for making us aware of the missing reference which was added to the revised manuscript.*

Section 2: A schematic of the models used would be beneficial to the reader.

*As reaction to your useful comment we added the models currently available within GLOFRIM to Figure 1 to not only provide textual, but also graphical information to the reader.*

Page 7, Line 37: Provide reference for the SRTM data.

*Thank you for making us aware of the missing reference which was added to the revised manuscript.*

Page 11, Line 7: Did you take into account uncertainties in the discharge at Obidos?

*Thank you for addressing this aspect. We neglected the uncertainty of observed discharge at Obidos as we assume that it is insignificant compared to other possible uncertainties, for instance parameterization of PCR-GLOBWB or surface roughness of the hydrodynamic models, particularly in large-scale modelling studies. As Clarke et al. (2000) reported an uncertainty of around 16 % of year-to-year variability, we added this reference and a brief comment to facilitate the reader's comprehension of model validation and its limitations.*

Page 11, Lines 14-20: Why are the results of the sensitivity analysis not included? The results are not surprising but you need to provide evidence.

*Thanks to your useful comment, an explanatory figure (Figure 6a and 6b) was added to the revised version of the manuscript, although we initially decided to not provide an additional plot as we assumed that this may distract the reader from the core of the manuscript. In addition to the plot, we extended the results section 5.2 accordingly, addressing the results and their implications. It is now possible to obtain a better idea of why variations in surface roughness and meandering coefficient can be excluded as cause for deviating discharge simulations between LISFLOOD-FP and Delft3D Flexible Mesh.*

Page 11, Line 25-26: What are the different gridding approaches applied? I'm not sure if this is stated elsewhere in the manuscript.

*To avoid any confusion, we named the gridding approaches explicitly. Thank you for pointing out this lack of clarity!*

*With the improvements made to the manuscript based on the valuable and critical reviewer's remarks, we are convinced to have responsibly addressed all ambiguities and shortcomings of the initially submitted version.*

**Rebuttal Letter Referee #2, Dai Yamazaki**

We thank referee #2, Dr. Dai Yamazaki, for the kind words on our manuscript and the points brought forward as they resulted in an improvement of the submitted manuscript. Below, we repeat the reviewer's comments, and provide our response in italics. In the revised manuscript, the changes made to the manuscript are highlighted in light blue.

5    P7. L32: "If RFS is activated, water volume is directly coupled to the 1-D channels of the hydrodynamic model while, when RFS is inactive, water is distributed over all grid cells of the 2-D domain": The description of RFS is not sufficient. Please explain the relationship between the hydrology model grid and hydrodynamic model pixels in a more detailed manner. I guess, "water volume of each coarse-resolution hydrology model grid" is distributed to the "corresponding high-resolution hydrodynamic model cells within the coarse-resolution grid". And the difference due to RFS is whether water volume is

10    distributed only to river cells or both river and floodplain cells within each coarse-resolution grid box. Readers who are not familiar with this topic might misunderstood water volume is distributed uniformly all-over the calculation domain (not to the corresponding cells).

*Thank you very much for pointing out the lack of clarity in our wording. Indeed, the River-Floodplain-Scheme functions as guessed by you. We thus re-wrote the explanation of the River-Floodplain-Scheme to avoid any ambiguity, and to improve*

15    *the reader's understanding of the coupling scheme.*

P7. L35: "the accuracy of the 2-D elevation data which is known to contain strong vertical bias, in particular when derived from remotely sensed global data": The elevation data is affected not only by vertical bias but also by various random/ systematic noises. I recommend to add reference to the latest research on this topic [Yamazaki et al., 2017].

*Since at the time of writing the current manuscript the MERIT DEM was not yet published, we could not refer to it despite*

20    *it's positive contribution to the current state of knowledge. However, we now added the reference to it in the revised manuscript.*

P8. L4: "We found that updating fluxes reduces run times compared to states, and hence advise opting for for this option": How downstream boundary conditions are treated. The two test cases executed in the manuscript assumes the downstream boundary is river mouth (0m constant). Some potential users might be interested to simulate flooding in middle-stream, that

25    requires a setting of downstream boundary conditions. Without a reasonable treatment of the downstream boundaries, it is difficult to state that the developed framework is "globally applicable". [I also note that "for" appears twice in the sentence.]

*We thank you for mentioning the missing information. It is, in fact, possible to employ any hydrodynamic schematization within GLOFRIM if it complies with Delft3D Flexible Mesh and LISFLOOD-FP requirements, respectively. This means that also other downstream boundaries besides constant water levels at a river mouth are feasible, for instance time-varying*

30    *water depths at a midstream observation point. For improved clarity, we extended and clarified the paragraph (section 5.1.) in the revised version of the manuscript.*

P8. L9: "although PCR runs in spherical coordinates": Given that a spherical coordination can be organized by a Cartesian system, it's better to clarify that PCR-GLOBWB runs at "non-Cartesian spherical system" while it is possible to use "regular lon-lat Cartesian system" for hydrodynamic models.

35    *Thank you very much for this comment. We updated section 3 accordingly.*

P8. L30. "4 The Synthetic Test Cases": It's better to explicitly state that PCR-GLOBWB is not used in the synthetic test case. I think this test case is done only for comparing Delft3D and LISFLOOD-FP under an ideal situation. Thus, this test case is directly not related to the GLOFRIM framework, thus readers might be confused.

*Based on your useful remark, we re-wrote the paragraph in the revised manuscript to improve clarity.*

40    P8. L34: "0.04 s m-1/3 for the 1-D run and 0.07 s m-1/3 for the 2-D run": Does this mean the same roughness coefficient is used for river channel and floodplains in 2D run? Usually, river channels have smaller roughness compared to floodplains.

*Thank you very much for your comment. It is worth mentioning that we performed a 1-D only and a 2-D only run, thus the latter did not contain any 1-D features. As this misunderstanding has also let to your remark regarding our choice of surface roughness values, we updated section 5.1 to increase clarity that the synthetic test case is really 2-D only in this case.*

P9. L25: "5.1 Set-up": Please describe how the downstream boundary was treated as this is critical for simulations. Please also explain how the complex channel network of the delta, bifurcating sections, and braided streams were treated. If they are treated differently by Delft3D and LISFLOOD-FP, this difference could be a potential cause of the disagreement of the simulation results.

*We thank you for mentioning this aspect. Since the schematization of LISFLOOD-FP is derived from Delft3D Flexible Mesh, both models employ the same 1-D network. In contrast to CaMa-Flood, the schematizations employed in the current manuscript do not account for bifurcations and thus the channel complexity of the delta had to be captured with only one channel. To acknowledge this shortcoming in our model schematizations as possible cause for deviations between simulated and observed discharge, we added information on not only the channel network in the delta, but also on how downstream boundaries were treated in section 5.1. Besides, we referred to it as possible source for the deviation of simulated and observed values in the discussion section 5.2.*

P10. L6. "for the elevation data the smoothed 5 canopy-free elevation data was upscaled to a 2 km spatial resolution": Please explicitly explain how the DEM was upscaled, because this has large impact on flood inundation. Did the authors took the mean within a cell, or the minimum elevation?

*As you rightly mention the influence of the chosen upscaling approach, we have extended the revised manuscript, noting that we used the nearest neighbour approach to avoid undesired smoothing effects in the elevation values.*

P10. L11. "roughness coefficient was uniformly set to 0.03 s m-1/3 for channel and floodplains": Was the same roughness used for channel and floodplain? If so, please clearly state, because different values are usually used.

*Thanks for your comment. Indeed, you are right that usually different surface roughness values are used for channel and floodplain, respectively. We, however, decided to use a uniform surface roughness value as this was also done by other studies as stated in the current manuscript. Hence, we desisted from further elaborating on this aspect in the revised manuscript.*

P10. L14: "For the hydrological model PCR-GLOBWB, the kinematic wave approach was used for routing outside of the coupled domain": Please explain that the simple kinematic routing may result in poor upstream boundary inflow, as backwater effects or river floodplain interactions can be neglected. Also, this approach could be a limitation for generating a realistic downstream boundary condition. Probably, using continental-scale hydrodynamic model (such as MGB-IPH or CaMa-Flood) as an intermediate step between the hydrology model and high-resolution hydrodynamic model can be a solution.

*We thankfully acknowledge your remark on the downsides of the kinematic wave approximation with respect to simulated upstream boundary flow computations. As you rightfully state, the upstream inflow signal can already deviate from observations due to the use of the kinematic wave approximation. Hence, we added this relevant aspect to the results section 5.2. Besides, we added your valuable proposition to employ as 1-D model such as MGB-IPH or CaMa-Flood to the concluding section 6.*

P10. L16: "we decided to apply a regionalized optimization technique": This "regionalized optimization" could be a limitation for using the proposed framework for "global application". The "modelling of flood inundation" may be possible at a global scale, but "global application" can be restricted by the quality of input/boundary datasets. Please state this limitation in the conclusion section.

*Thanks for pointing out that a global application can be locally restricted by data availability for model set-up. In the revised manuscript, we pointed out that the "regionalized optimization" is optional and PCR-GLOBWB can also be run as is in its default parameterization, thus not posing any constraint to a global application. We described both aspects in more detail in the description of the set-up (section 5.1) to avoid any ambiguity in this matter.*

5  P11. L6: "5.2 Results and Discussion": More detailed analysis of the difference between Delft3D and LISFLOOD-FP is needed. As far as I guess from the figures, the flood peak of Delft3D is later than LISFLOOD-FP because it has larger inundation in upstream areas due to its coarser flexible cell resolution. The smaller water level amplitude in upstream must be also related to the larger inundation in upstream. Because floodplain inundation attenuate flood waves, suppress water level fluctuations, and delays flood peak, most of the disagreement between the two models can be explained consistently

10  due to the inundation in upstream regions. The authors can analyze this effect easily, by comparing the simulated discharges by Delft3D and LISFLOOD-FP also at upstream locations other than the Obidos. By analyzing discharge, we can show where flood waves were attenuated. I suggest to include this discussion in the manuscript. And if the discussion above is true, the Delft3D simulation must be sensitive also to the spatial resolution in upstream regions, thus I recommend to include some sensitivity test on the spatial resolutions.

15  *We are thankful for your extended remark on the model results. First, we agree with the assumption made that the spatial resolution applied by Delft3D Flexible Mesh in upstream areas may impact model results locally as well as further downstream. To shed light on this issue, we added a comparison of simulated discharge for two stations upstream of Obidos to obtain a first-order impression (Figure 4 and Figure 6c). Because we are currently working on a follow-up study concerning the relation of spatial resolution of flexible meshes and model results, we desisted from providing a too elaborate*

20  *discussion in the current version of the manuscript and added a limited discourse only to section 5.2. Besides, we recommend further investigation of this linkage between cell size resolution and simulated discharge in section 6.*

P11. L16: "Since the routing scheme of LFP is based on a D4 system, channel length and dimension in LPF tend to be longer than in other hydrodynamic models": This is not precise. The D4 river network can generate shorter channel length if the scale of channel meandering is smaller than the size of the cell. Please rewrite this sentence. Furthermore, the D4 system

25  does not only change the flow length. It alters the connectivity of channels and floodplains. Some channels/floodplains which are connected in the D8 system (or a vector system) could be disconnected in the D4 system, because diagonal connectivity is not allowed. To avoid this problem, the DEM should be adjusted to ensure the D4 connectivity. Please make a discussion on this issue, as this could be one of the main reason of the difference between Delft3D and LISFLOOD-FP.

*Thank you for the information on the D4 river network system and the related uncertainties. In the revised manuscript, we*

30  *extended the description of the D4 system and its limitations. To show the impact of both increasing and decreasing channel dimension, we furthermore added a plot of the results from the conducted sensitivity analysis (Figure 6a) to better supplement our results and discussion section 5.2 which we updated accordingly, even with the results not indicating any notable change in discharge with varied meandering coefficients in LISFLOOD-FP.*

P11. L29: "While at the most upstream station Loc3 DFM simulates lower water levels than LFP": This is probably due to

35  larger flooding in Delft3D due to its coarser spatial resolution in upstream, as discussed above. Please clarify.

*Thank you for this remark. We agree with your suggestion and have added the link between differences in simulated water level and simulated inundation extent to the revised manuscript. Besides, we qualitatively correlae it to the additional discharge simulations made in upstream areas.*

P11. L34: "the more pronounced difference in water levels at Loc1 may simply be a local effect": What is the "local effect".

40  Please explain in detail.

*Thank you for pointing out the lack of clarity. A previous study already showed that the behaviour of simulated water level is not always predictable due to spatial feedback dynamics between neighbouring cells of an observation station (Hardy et al., 1999). Since we cannot really explain the more pronounced difference in water levels at this location with our current process understanding of the coupled set-up, we added this reference to the discussion of model results in the revised*

5 *manuscript as a possible source of error.*

*With the additions made to the manuscript based on the valuable and critical reviewer's remarks, we are convinced to have responsibly addressed all uncertainties, ambiguities, and shortcomings of the initially submitted version.*

**Main Changes to Manuscript**

Comments reviewer #1 (changes made highlighted yellow in revised manuscript):

1. Replaced too colloquial language in the manuscript.
2. Provided spatial resolution of the CRU-data as well as link to document describing the preparation of CRU-forcing in PCR-GLOBWB in section 2.1.
3. Added the models currently available within GLOFRIM to Figure 1.
4. Added reference to the original Shuttle Rader Topography Mission (SRTM) data to section 3.
5. Mentioned possible uncertainty in observed discharge data, and provided source quantifying this uncertainty in section 5.1.
6. Plotted results of sensitivity analysis of both meandering coefficient and surface roughness in LISFLOOD-FP in Figures 6a and 6b, respectively, and elaborated on it section 5.2.
7. Explicitly re-stated the different gridding techniques employed by the two hydrodynamic models in section 5.2.

Comments reviewer #2 (changes made highlighted light-blue in revised manuscript):

1. Re-wrote the description of the River-Floodplain-Scheme in section 3.
2. Added reference to the MERIT DEM in section 3.
3. Updated and re-wrote section to clearer describe functionality of GLOFRIM with respect to the use of different types of downstream boundaries in section 5.1.
4. Explicitly stated that PCR-GLOBWB is not made use of in the synthetic test case in section 4.1
5. Clearly stated how downstream boundaries, river braiding, and bifurcations are treated in the hydrodynamic schematizations used in the study in section 5.1, and discussed possible shortcomings in the schematization with respect to model results in section 5.2.
6. Added the upscaling technique applied and gave reasoning in section 5.1.
7. Did not made changes to manuscript since reasoning and references for using a uniform surface roughness coefficient are already provided in manuscript
8. Mentioned use of kinematic wave approximation as potential source of error in section 5.1, and elaborated on it accordingly in discussion section 5.2; also, recommended use of large-scale 1-D models for upstream/midstream section in section 6.
9. Stated in section 5.1 more clearly that regional optimization is only optional and advised for catchment studies, hence not contradicting any global application.
10. Added two stations upstream of Obidos (see updated Figure 4) and assessed simulated discharge; added discussion of results to section 5.2 and recommended further investigation in section 6.
11. Improved description of D4 system and assessed impact of accounting for both over- and underestimation of channel dimension in Figure 6a; briefly elaborated on it in section 5.2.
12. Established stronger relation between spatial resolution and simulated water levels in section 5.2.; besides, used additional upstream discharge simulation to underpin the statement.
13. Added explanation and reference of the local effects that can be observed for water level simulations in section 5.2.

[revised manuscript text omitted]